# Epigenetic Restriction Factors (eRFs) in Virus Infection

**DOI:** 10.3390/v16020183

**Published:** 2024-01-25

**Authors:** Arunava Roy, Anandita Ghosh

**Affiliations:** Department of Molecular Medicine, University of South Florida, Tampa, FL 33612, USA; anandita1@usf.edu

**Keywords:** epigenetic viral restriction factor (eRF), viral chromatin, PML-NB, KRAB/KAP1, IFI16, HUSH complex, ADAR, pseudouridine synthases (PUS), N6 methyl adenosine (m6A), epitranscriptomics

## Abstract

The ongoing arms race between viruses and their hosts is constantly evolving. One of the ways in which cells defend themselves against invading viruses is by using restriction factors (RFs), which are cell-intrinsic antiviral mechanisms that block viral replication and transcription. Recent research has identified a specific group of RFs that belong to the cellular epigenetic machinery and are able to restrict the gene expression of certain viruses. These RFs can be referred to as epigenetic restriction factors or eRFs. In this review, eRFs have been classified into two categories. The first category includes eRFs that target viral chromatin. So far, the identified eRFs in this category include the PML-NBs, the KRAB/KAP1 complex, IFI16, and the HUSH complex. The second category includes eRFs that target viral RNA or, more specifically, the viral epitranscriptome. These epitranscriptomic eRFs have been further classified into two types: those that edit RNA bases—adenosine deaminase acting on RNA (ADAR) and pseudouridine synthases (PUS), and those that covalently modify viral RNA—the N6-methyladenosine (m6A) writers, readers, and erasers. We delve into the molecular machinery of eRFs, their role in limiting various viruses, and the mechanisms by which viruses have evolved to counteract them. We also examine the crosstalk between different eRFs, including the common effectors that connect them. Finally, we explore the potential for new discoveries in the realm of epigenetic networks that restrict viral gene expression, as well as the future research directions in this area.

## 1. Introduction

Viruses can infect all forms of life, and almost always, such encounters are detrimental to the host. To counteract viral infections and protect themselves and their genetic integrity, vertebrates have evolved various antiviral immune responses. Antiviral immunity can be broadly classified into two categories: the innate immune responses and the adaptive immune responses. The innate immune responses are the first line of defense against invading viruses and typically comprise the cellular interferon and inflammasome pathways, which produce type I and III interferons and interleukins (IL)-1β and IL-18, respectively [1]. The adaptive immune responses, which constitute the humoral and cell-mediated responses, are the second line of systemic defenses, which kick in after the virus has successfully breached the innate immune barriers [2]. However, both the innate interferon and inflammasome pathways as well as the adaptive immune system need to be invoked via intricate signaling cascades, which makes their responses delayed. Fortunately, a more immediate, cell-intrinsic, stand-alone antiviral mechanism, called the antiviral restriction factors, or RFs, exists [3]. These are host proteins that recognize and restrict specific viral proteins or processes to limit viral transcription and replication without any upstream signaling events and are, therefore, the actual first line of antiviral defenses that engage even before the interferon and inflammasome pathways. Because of this major difference, RFs, though originally considered a part of the innate immune systems, are now increasingly being considered as the third arm of antiviral immunity [4]. Additionally, in contrast to innate immune responses, which are not specific to a pathogen and recognize broad-spectrum viral pathogen-associated molecular patterns (PAMPs) via cognate pattern-recognition receptors (PRRs), RFs are often more specific to a particular group of viruses and interfere with a particular viral protein or process [4]. Most RFs are constitutively expressed in different cell types and are thus readily available to counter viral infections; however, some RF genes are interferon-stimulated genes (ISGs), whose expression is further augmented upon interferon stimulation. In recent years, several RFs with diverse mechanisms of action have been identified [3]. For example, APOBEC3G is a cytidine deaminase that inhibits retroviral reverse transcription, MxA protein is a GTPase that targets influenza A virus ribonucleoproteins (vRNPs) to inhibit replication, and TRIM23 is an E3 ubiquitin ligase that targets HSV-1 replication [5,6,7]. Among these diverse mechanisms of viral restrictions, a discrete group of RFs imparts their restriction function by modifying the viral epigenome. In this review, these RFs are termed ‘epigenetic restriction factors’ or eRFs.

Epigenetics defines a branch of gene regulation that is beyond the genetic code itself. Eukaryotic gene expression is not only regulated by the presence of cis-acting promoters and enhancer elements but also by a variety of epigenetic machinery that direct the recruitment and accessibility of transcription factors, RNA stability, and translation efficiencies. Among these are (i) chromatin modifications such as DNA methylation, histone modifications, nucleosome remodeling, and alternative histone variant usage, all of which alter DNA accessibility and/or chromatin structure, and (ii) post-transcriptional RNA modifications such as RNA adenine methylation and cytosine acetylation that alter RNA structure, stability, translation efficiency, etc. [8]. Epigenetic modifications can either positively or negatively regulate gene expression. Being obligate parasites, viruses hijack almost every aspect of their host’s molecular machinery, including its epigenetic machinery, to successfully orchestrate their lifecycle. This also makes them vulnerable to the same gene-regulatory epigenetic processes as the host’s chromatin and mRNAs. As viruses and their hosts co-evolved, host cells developed specific negative epigenetic gene regulatory mechanisms to recognize and restrict viral transcription and translation. These eRFs are now recognized as important players in cellular defense against both DNA and RNA viruses.

This article comprehensively reviews the eRFs identified so far and discusses their mechanism of restriction, interacting partners, and effect on viral transcription and replication. Some viruses have also evolved specific mechanisms to counteract their cognate RFs to ensure their survival after infection. Consequently, there is a constant evolution of RFs and their viral antagonists as they engage in a never-ending arms race. We will also explore these eRF antagonizing viral mechanisms. Intriguingly, recent research suggests that some viruses have evolved to utilize specific eRF-mediated gene silencing mechanisms to establish a state of intercellular dormancy called latency, which helps them to persist until appropriate conditions arise for them to reactivate and complete their infectious cycle. We will explore examples of such exploitation of eRFs by viruses.

The state of viral latency, which is capable of reactivation to produce infectious virion particles, is different from another form of retroviral genetic material persistence in mammalian cells, called endogenous retroviruses (ERVs). ERVs are a type of retrotransposons called long terminal repeat (LTR) retrotransposons that are remnants of once infectious exogenous retroviruses that integrated into germline cells of mammals and were retained through vertical inheritance [9]. They constitute as much as 8% of the human genome [10]. ERVs are classified into three classes (I, II, III) based on the similarity of their reverse transcriptase genes or their relationship to different genera of exogenous retroviruses [11]. ERVs play important roles in human health and disease, such as placenta development and immune defense [12,13]. Although most endogenous retroviruses (ERVs) are not latent viruses in the true sense and are unable to replicate or reactivate to produce infectious virions in humans [13], there are some rare examples of ERVs that can produce infectious viruses, such as the feline ERV known as RD-114 [14]. Also, some copies of murine ERVs, like intracisternal A-particle (IAP) and MusD sequences, are still considered transposition-competent as they can be transcribed into RNA, reverse transcribed into DNA in the cytoplasm, and then integrated into the genome at a new location. In mammals, ERVs are subject to tight transcriptional repression in adult tissues through the action of several gene-silencing mechanisms that restrict their expression, an abundance of which could potentially trigger an autoimmune response. Here, we will also discuss the host restriction mechanisms that aid in silencing ERFs. 

Epitranscriptomics is the epigenetic regulation of nascent RNA. Numerous covalent RNA modifications have been discovered in eukaryotes, but only a handful of them have been observed in viral RNAs. We will discuss those epitranscriptomic modifications that have been identified to serve as antiviral restriction factors.

## 2. Virus Epigenetic Restriction Factors (eRFs)

This article categorizes eRFs into two classes based on the type of epigenetic machinery they use. The first class consists of eRFs that target viral chromatin and restrict DNA viruses and retroviruses by depositing epigenetic silencing marks on their chromatin, thereby inhibiting their transcription. The second class consists of eRFs that modulate RNA post-transcriptional modifications and restrict mRNA translation of both DNA and RNA viruses. These eRFs have other cellular functions, including host gene regulation, immune system modulation, and cellular response to non-viral invasions. However, in this article, we will only focus on their epigenetic antiviral restriction functions.

### 2.1. eRFs That Target Viral Chromatin

Chromatin-modulating eRFs neutralize invading DNA threats like nuclear-replicating DNA virus genomes, retroviral-RNA-derived DNA elements, and ERVs by depositing epigenetic silencing marks like H3K9me3 and H3K27me3 to restrict their transcription [8,15,16,17]. To date, four chromatin-targeting eRFs have been recognized (Figure 1)—the promyelocytic leukemia nuclear bodies (PML-NBs or ND10), the Krüppel-associated box protein (KRAB)/Krüppel-associated box domain-associated protein 1 (KAP1 aka TRIM28) complex, interferon-inducible protein 16 (IFI16), and the human silencing hub (HUSH) complex (Figure 1). Many other members of the cellular heterochromatin-establishing machinery, such as the EZH2 complex, various H3K9methyltransferases (H3K9 MTases), histone deacetylases (HDACs), DNA methyltransferase (DNMT), etc., also play active roles in the restriction of a wide variety of DNA viruses. However, they are not individually regarded as eRFs because they need to be recruited onto the invading viral DNA by other proteins or protein complexes, which recognize the foreign DNA and provide a macromolecular scaffold for the assembly of a concerted heterochromatization machinery. Thus, here, we will consider these protein complexes as chromatin-targeting eRFs.

#### 2.1.1. PML-NBs or ND10

The promyelocytic leukemia nuclear bodies (PML-NBs) are dynamic, membrane-less subnuclear domains, 0.1–1.0 µm in diameter, found in most cell lines and many tissues [18,19]. They are also referred to as the nuclear domains 10 (ND10) based on their average number of 10 loci per nucleus in various cultured cells [20]. They are a multiprotein complex consisting of a diverse and dynamic array of both positive and negative transcription regulators known to be involved in many key cellular processes, including DNA damage response, apoptosis, autophagy, cellular differentiation, oncogenesis, epigenetic silencing, and antiviral defenses [18,19,21,22,23]. In addition to the PML-NBs’ permanent members, like PML (TRIM19), Sp100, and DAXX, various other proteins transiently reside in the PML-NBs based on cell stage, type, and condition. DAXX and its binding partner, ATRX, together form a histone chaperone complex, which recruits the non-canonical histone variant, H3.3, to the target loci [24]. It also recruits epigenetic modifiers such as HP1 and the histone methyltransferase EZH2 which is responsible for the deposition of H3K27me3 marks [25]. As many as 271 proteins have been reported to be associated with PML-NB under various conditions [22]. Many of these proteins have been recognized as intrinsic eRFs in diverse virus systems, but because they all colocalize with the PML-NBs, we will collectively consider them as part of the same macromolecular ND-10 complex. 

Several recent observations establish that the PML-NBs serve as a subnuclear hub for the epigenetic repression of the DNA of several viruses. The juxtaposition of the latent HIV provirus to the PML-NB was found to be responsible for the recruitment of G9a, an H3K9 methyltransferase, responsible for the deposition of H3K9 dimethylation (H3K9me2) on the provirus promoter [19,26]. Knockdown (KD) of PML led to a decrease in provirus bound G9a, loss of H3K9me3 heterochromatin marks, and gain in the euchromatic mark, H3K4me3, on the HIV promoters [26]. In the case of ERVs, retrotransposition of the mouse IAP and MusD elements were found to be potently inhibited by the PML-associated DAXX protein [27]. In herpes simplex virus (HSV-1), PML-NBs have been reported to recruit the H3.3 chaperone complexes, DAXX-ATRX and HIRA (HIRA, UBN1, CABIN1, and ASF1a), leading to the viral genomes being heterochromatinized, almost exclusively with H3.3K9me3 [19,28]. ATRX has also been reported to promote the maintenance of the heterochromatin on the HSV-1 genome [29]. In another study, ATRX was found to restrict the accessibility of the HSV-1 chromatin to the transcription machinery [30]. Similarly, in another herpesvirus, the human cytomegalovirus (HCMV), several reports have established the role of PML-NBs and their associated factors like Sp100 and DAXX in epigenomic gene silencing. DAXX has been demonstrated to silence the expression of HCMV immediate-early (IE) genes by recruiting the chromatin-remodeling protein ATRX or chromatin-modifying enzymes like histone deacetylases (HDACs) or promoting H3.3 deposition to the viral DNA leading to heterochromatinization of its major immediate-early enhancer/promoter (MIEP) [31,32,33,34,35,36,37]. In the Kaposi sarcoma-associated herpesvirus (KSHV), KD experiments in de novo infected primary cells showed that the virus infection is restricted by the ND10 components PML and Sp100 but not by ATRX [38]. In adenoviruses (AdVs), it has been reported that DAXX changes the epigenetic status of viral promoters in cooperation with ATRX by repositioning H3.3 variants [39]. Also, Sp100A, a PML-NB protein, recruits the heterochromatin protein 1 alpha (HP1α) to condense viral DNA, leading to transcriptional repression of the AdV genome [22]. FOXO4, a PML-NB-associated transcription factor, was found to promote heterochromatinization and silencing of hepatitis B virus (HBV) DNA [40]. In addition, the structural maintenance of chromosome 5 and 6 (Smc5/6) proteins, which also localize with PML-NBs in human hepatocytes, has been reported to act as RFs for HBV transcription by altering their chromatin organization [34,41,42]. However, no DNA sequence specificity of any PML-NB components has been discovered. 

Many viruses, specifically nuclear-replicating DNA viruses, must antagonize PML-NB-induced heterochromatinization to establish a productive infection. AdV E1A-13S protein, HCMV pp71 protein, HBV HBx protein, EBV BNRF1, KSHV ORF75, and HSV-1 ICP0 protein all induce the degradation, disruption, or reorganization of PML-NB-components, resulting in the disassembly of the structures, which is required to initiate viral transcription and replication [34,35,38,43,44,45,46,47,48]. Apart from ubiquitin-mediated degradation and protein–protein interaction disruption, DNA viruses have evolved another potent way to antagonize the PML-NBs. SUMOylation of PML is essential for the integrity of PML-NBs. Viral proteins such as the HCMV IE1 and KSHV vIRF3 de-SUMOylate the PML protein, causing a complete dispersal of the nuclear bodies [49,50].

On the contrary, some viruses that establish long-term latency in their hosts have been proposed to exploit the PML-NB-mediated epigenetic silencing to aid in their latency establishment and maintenance. For example, HCMV utilizes the PML-NB-mediated heterochromatinization of its MIEP to attain a silenced chromatin structure in latently infected cells [51]. In mouse models of HSV-1 infection, the viral genome localized at PML-NBs during acute phases of infection and subsequent latent phases in neurons, and the depletion of PML affected the expression of latency-associated transcripts (LAT) and the subsequent establishment of latency [52]. However, contrasting observations exist in other herpesviruses, including KSHV and EBV, where the establishment of latency was found to be independent of PML proteins [53,54,55]. Thus, it is unlikely that a common mechanism of interplay between PML-NBs and viral genomes during latency establishment and maintenance exists for all DNA viruses. More research is needed to comprehensively understand the complex interplay between the PML-NB-associated proteins and the epigenetic modulation of herpesviral genomes. In human papillomavirus (HPV), it has been discovered that the PML protein is important in retaining incoming viral DNAs in the nucleus and for the transcription of viral genes [56]. However, no epigenetic mechanism has been found to explain the dependence of HPV on the PML-NBs.

#### 2.1.2. KRAB/KAP1 Complex

KRAB (Krüppel-associated box) domain-containing zinc-finger proteins (KRAB-ZFPs) and their corepressor, KRAB-associated protein 1 (KAP1 aka TRIM28 or TIF1β), regulate mammalian transcription through epigenetic marks and chromatin compaction [57,58]. KAP1 is a tripartite-motif (TRIM) protein recruited to DNA via interaction with various KRAB-ZNF (zinc-finger) proteins, which are a family of sequence-specific DNA-binding proteins. The sequence specificity of KRAB-ZFPs is mainly determined by three key specificity-determining amino acid residues within each zinc finger (ZF) [59]. The identity of these amino acids and the number of tandem ZFs in a particular KRAB protein determine its sequence specificity. Like the PML-NBs, the KRAB/KAP1 system is also a dynamic macromolecular complex, which, upon tethering to chromatin, serves as a scaffold to recruit various epigenetic repressive machineries, such as histone deacetylases (e.g., NuRD) [60], H3K9 methyltransferases (e.g., SETDB1) [58], and heterochromatin-protein 1 (HP1) proteins [61,62], to promote chromatin condensation and transcriptional repression. Apart from virus restriction, KAP1-mediated epigenetic modulations have been reported to regulate multiple physiological processes, including cell differentiation, DNA damage response (DDR), immune response, and tumorigenesis [63]. Apart from epigenetic regulation, KAP1 also serves non-epigenetic functions such as a SUMO/ubiquitin E3 ligase and a signaling scaffold for mediating signal transduction [63]. Such non-epigenetic functions of KAP1 have also been identified in viral restriction. For example, KAP1 has been reported to promote the proteasomal degradation of the HIV transactivator protein, Tat, which restricts viral replication [64].

As an eRF, KAP1 has been reported to epigenetically restrict the HIV genome by serving as an adaptor protein recruiting HP1, SETDB1, and the NuRD complex [65]. Upon KD of KAP1, a significant decrease in H3K9me2, H3K9me3, and H3K27me3, as well as a substantial increase in H3K4me3 and H3K9Ac, were observed [64,65]. Similarly, in other retroviruses, such as the murine leukemia virus (MLV) and the prototype foamy virus (PFV), KAP1-mediated H3K9me3 and HP1 deposition have been reported [15,66,67]. KAP1 deletion also led to a marked upregulation of the mouse IAP ERV in embryonic stem cells and early embryos due to the loss of H3K9me3, proving that KAP1 also controls endogenous retroelements [68]. In AdVs, KAP1 has been reported to restrict viral transcription by associating with SPOC1 (SPOC domain-containing 1), a chromatin-remodeling factor that has been reported to induce the deposition of H3K9me3 marks [69]. A similar mechanism of HCMV restriction has emerged, where KAP1-associated SPOC1 impaired HCMV replication by binding to the HCMV major immediate-early promoter (MIEP) and subsequently recruiting other heterochromatin-building factors [70]. Herpesviruses such as EBV, KSHV, and HSV1 have all been shown to be restricted by KAP1 via H3K9me3 deposition [71,72,73,74]. In KSHV, it has been reported that the latency-associated nuclear antigen (LANA) mediates KAP1 recruitment to the ORF50 promoter, resulting in repressive epigenetic modifications by decreasing activating histone acetylation (AcH3), possibly through recruitment of the histone deacetylase complexes HDACs 1 and 2 [75]. In line with this, we have previously reported that LANA facilitates KAP1’s interaction with another host transcription factor, Nrf2 (Nuclear factor E2-related factor 2), which together represses viral lytic gene expression [76]. In adeno-associated virus (AAV), KAP1 was found to bind to the latent genome at the rep ORF, leading to H3K9me3 deposition and heterochromatization [77].

HIV-1 and MLV have been reported to utilize KAP1-mediated epigenetic silencing to promote their latency maintenance [65,66]. Herpesviruses like EBV and KSHV also exploit the KAP1-mediated recruitment of epigenetic repressive machinery and the subsequent compaction of viral chromatin for latency maintenance [72]. During de novo infection of KSHV, LANA-mediated recruitment of KAP1 and its downstream epigenetic silencers aid in the shutdown of lytic gene expression during the early stage of KSHV, facilitating latency establishment [75]. In AAV, which needs a helper virus for replication, KAP1-mediated heterochromatin deposition was found to be essential for latency establishment and maintenance [77]. When reactivation is needed, the AAV Rep protein antagonizes KAP1 by recruiting the protein phosphatase 1 (PP1) to phosphorylate KAP1 at S824 (p-KAP1-S824). This results in the release of the repressive complex, relaxation of heterochromatin, and relief of transcriptional repression [77]. Another mechanism of KAP1 antagonism has emerged in AdV, where HAdV5-induced KAP1 deSUMOylation is known to promote chromatin decondensation, thus overcoming KAP1-mediated epigenetic gene silencing [69].

#### 2.1.3. IFI16

The interferon-gamma inducible protein 16 (IFI16) is an IFN-responsive PYHIN family DNA-binding protein that performs various essential roles in a wide range of cellular activities. These include roles in DNA damage response, myeloid cell differentiation, pyroptosis, transcription regulation, and as an antiviral restriction factor [78,79,80,81,82,83,84,85]. In its role as an antiviral RF, IFI16 is a multifaceted protein with diverse molecular functions. These include sensing of viral DNA and induction of the IFN-1 [86,87], and inflammasome cascades [78,81], sensing of viral RNA and induction of IFN-1 [88], and pyroptosis pathways [82], transcription regulation of viral gene expression, and epigenetic silencing of viral nucleic acids [89,90,91,92,93,94]. As a sensor of foreign nucleic acid, which is probably its most prominent antiviral role, IFI16 has been reported to restrict a number of DNA viruses like HSV-1 [89,91,92], KSHV [81,87], EBV [95], HCMV [96,97], and HPV18 [90], and a number of RNA viruses like influenza virus (IAV) [88], Chikungunya virus (CHIKV), Zika virus [98], and HIV-1 [99]. In its transcription regulator role, IFI16 has been reported to restrict HIV-1 and HCMV by targeting the transcription factor Sp1 [96,100] and HSV-1 by occluding RNA pol II, TATA-binding protein (TBP), and the transcription factor Oct1 (organic cation transporter 1) [89].

However, distinct from these DNA/RNA sensing and transcription factor-modulating roles, IFI16 also mediates viral gene restriction by epigenetically modifying the viral chromatin landscape. Several initial reports suggested IFI16’s epigenetic role in the regulation of herpesviral genome DNA. IFI16 was observed to promote the recruitment of heterochromatin marks while concomitantly reducing euchromatin marks on the chromatin of an ICP0 null HSV-1 and HPV [89,90,92]. Subsequently, it was reported that IFI16 forms filamentous structures on ICP0 null HSV-1 chromatin, which in turn, drive the deposition of H3K9me3 heterochromatin mark by 6 h post-infection (hpi) [91]. Following this, we have demonstrated that IFI16 interacts with and recruits the H3K9 MTases SUV39H1 (suppressor of variegation 3–9 homolog 1) and GLP (G9a-like protein) on KSHV lytic genes, leading to the enrichment of H3K9me3 marks [94]. This leads to the recruitment of the chromatin compactor protein, HP1α (heterochromatin protein 1α), which eventually silences lytic genes [94]. Another report observed that upon IFN stimulation, IFI16 helps in stabilizing the H3K9me3 heterochromatin on the HSV-1 genome [101]. The binding of IFI16 to the HSV-1 genome has been found to be sequence-independent, with the highest binding occurring at the most accessible regions of the viral chromatin [102]. A similar epigenetic silencing role of IFI16 has emerged in HBV, where it promotes the epigenetic suppression of HBV covalently closed circular DNA (cccDNA) by targeting an interferon-stimulated response element (ISRE) present in cccDNA [103]. However, the role of IFI16 in the epigenetic regulation of other DNA viruses commonly restricted via the epigenetic route, like AdV and AAV, has not yet been reported.

Viruses have evolved to counter the repressive effects of IFI16. ICP0 protein of HSV-1 promotes the degradation of IFI16 in a RING domain- and proteasome-dependent manner [104]. Our previous observations showed that in KSHV, IFI16 is degraded via the proteasomal pathway during latency to lytic switch [93]. We also made similar observations in EBV, where an immediate-early or early viral protein was found to be responsible for the reduction of steady-state IFI16 protein levels post-reactivation from latency [95]. However, the viral protein responsible for these degradations has not been identified yet. In HCMV, the pUL83 tegument protein (pp65) has been reported to bind to the pyrin domain of IFI16 and prevent its multimerization [105]. Though this has been shown to prevent initiation of the IRF-3 signaling pathway, whether it has a role in inhibiting IFI16 nuclear filaments and their downstream epigenetic silencing is not yet known. The HPV E7 protein has been reported to interact with IFI16 and promote its TRIM21-mediated proteasomal degradation [106]. This loss of IFI16 has been observed to inhibit cell pyroptosis through the reduction of IFI16 inflammasomes. However, whether this mechanism of IFI16 degradation also counters IFI16-mediated heterochromatinization of the HPV genome needs to be investigated. The eRF functions of IFI16 help in the establishment and maintenance of latency in herpesviruses. In KSHV, we showed that IFI16 is vital for the establishment and maintenance of latency [93], and KD of IFI16 results in loss of heterochromatic marks and subsequent lytic reactivation [94]. Similarly, we also showed that in EBV, IFI16 is essential for the maintenance of latency [95]. Another report showed that IFI16 partners with KAP1 to maintain EBV latency [74]. It is worth noting that IFI16 links the innate immunity network to the epigenetic regulatory system. IFI16, being an IFN-inducible gene, relays autocrine and paracrine IFN signals to the nuclear heterochromatinizing machinery to rapidly ramp up its virus restriction function [101].

#### 2.1.4. The HUSH Complex

The human silencing hub (HUSH) complex is a heterotrimer composed of TASOR (transcription activation suppressor), MPP8 (M-phase phosphoprotein 8), and PPHLN1 (periphilin). It recruits two heterochromatinizing effectors, MORC2 (microrchidia CW-type zinc finger 2), an ATP-dependent chromatin remodeler that compacts chromatin, and SETDB1 (SET Domain Bifurcated Histone Lysine Methyltransferase 1), a histone methyltransferase (MTase) that deposits repressive H3K9me3 marks at the target loci [107,108]. Another protein, ATF7IP (activating transcription factor 7 interacting protein), mediates the stabilization of SETDB1 and is considered essential for heterochromatin formation by the HUSH complex [109]. The HUSH complex localizes to H3K9me3-rich regions on chromatin in a sequence-independent manner and induces the spread of heterochromatin. It plays many important roles, including embryonic development [110,111,112], position-effect variegation of gene expression [113], suppression of inappropriate immune and interferon activation [114,115], control of brain architecture [116], and the transcriptional silencing of foreign RNA-derived DNA elements inside the nucleus, including long terminal repeat (LTR) retrotransposons like ERVs and non-LTR retrotransposons like LINE-1 elements [117,118]. 

The first indication that the HUSH complex has an eRF role was discovered using a spleen focus-forming virus (SFFV, a mouse retrovirus) promoter-driven integration-competent reporter construct [117]. The same study also reported the HUSH-dependent silencing of integrated HIV-1 and MLV reporter viruses. Subsequently, the HUSH-mediated silencing of unintegrated retroviral DNA was reported, and the DNA-binding protein NP220 was shown to be instrumental in recruiting the HUSH complex onto the viral DNA [119]. Similar observations were also made using integration-deficient MLVs where NP220 recruited the HUSH components and SETDB1 to silence the viral genome by depositing H3K9me3 marks [119]. HUSH’s role in the restriction of ERVs was established by the demonstration that loss of HUSH activity via SETDB1 KO conferred de-repression of ERVs in melanoma cells [120]. Another study has reported that the HUSH component, FAM208A, binds to ERV DNAs [121]. Apart from retroviruses and retroviral elements, another genome-integrating virus of the parvovirus family has been reported to be sensitive to HUSH-mediated viral restriction. A recombinant adeno-associated virus (AAV) containing a transgene has been observed to be susceptible to NP220-driven HUSH complex silencing [17,122].

The HUSH complex has been reported to assist in the persistence of latent HIV-1 proviruses in CD4+ memory T cells by limiting provirus transcription so that the host immune system cannot detect the presence of the virus [123]. In support of this hypothesis, it has been found that the primate immunodeficiency viruses’ accessory proteins, Vpx and Vpr, associate with the HUSH complex and decrease the steady-state level of these proteins in a proteasome-dependent manner, thus resulting in the activation of the integrated proviruses [123]. Similar observations have also been made in HIV-2 and SIV, where the Vpx protein induces the degradation of TASOR to eventually downregulate the HUSH complex [124,125,126]. As a result, H3K9me3 marks depletion, and reactivation of latent HIV proviruses was observed [124]. However, mutant Vpx that was unable to induce HUSH degradation did not cause reactivation [124].

### 2.2. Epitranscriptomic eRFs

Epitranscriptomic modification of RNA is a rapidly expanding field, and about 170 different covalent modifications have been identified that modify mRNAs, tRNAs, and non-coding RNAs (ncRNAs) at the single-nucleotide level [127,128]. Among these are post-transcriptional RNA modifications such as nucleotide methylation and acetylation and post-transcriptional RNA editing processes such as nucleotide insertion, deletion, or substitution. These modifications and edits directly or indirectly regulate RNA structure, stability, localization, alternative splicing, nuclear export, translation, microRNA biogenesis, and immune recognition [129]. Out of the 100+ epitranscriptomic RNA modifications identified in mammals, only four types have been reported to impact viral gene expression: N6-methyladenosine (m6A), 5-methylcytidine (m5C), N4-acetylcytidine (ac4C), and 2′O-methylation of the ribose moiety of all four ribonucleosides (collectively Nm) [129]. Though all these RNA modifications have prominent proviral roles, only one among these, m6A, has been reported to have viral restriction functions (Figure 2). Apart from these, two types of posttranslational RNA editing systems have been reported to edit viral RNA: adenosine to inosine editing and uridine to pseudouridine editing (Figure 2) [129]. Both have recognized roles in viral restriction (Figure 2). Apart from the editing or modification of viral RNAs, several examples exist in which the epitranscriptomic modification of cellular RNAs, particularly those involved in the IFN pathways, has been shown to have antiviral effects on many viruses. However, this review will only discuss the antiviral outcomes of epitranscriptomic modification of viral RNA and not cellular RNAs.

#### 2.2.1. Adenosine Deaminase Acting on RNA (ADAR)

ADAR catalyzes adenosine to inosine (A to I) editing in dsRNA by deaminating adenine at C6 to produce inosine, which is chemically identical to guanosine (G) with the exception of the absence of the amino group attached to C2 (Figure 2) [130]. As a result, converting A residues to I can alter the coding capacity of the ribosome or viral RNA-dependent RNA polymerases (RdRp), change splice sites, or alter the structure of RNA. There are three ADAR enzymes (ADAR1-3) in mammalian cells; however, ADAR3 has not yet been shown to have deaminase activity [131]. ADAR1 has two isoforms: p110, which is constitutively expressed and localizes in the nucleus and p150, which is induced by IFN signaling and localizes in both the cytoplasm and nucleus. ADAR2, on the other hand, is constitutively expressed [131]. ADARs play many important physiological functions including innate immunity, RNA splicing, RNA interference, protein recoding, neural functions, autoimmunity, embryonic development, cell differentiation, immune regulation, and viral restrictions [132].

Although ADAR plays a significant role in regulating various molecular processes of many different viruses, many of these roles are proviral and assist the virus biology on numerous levels [129,133,134]. Nonetheless, multiple reports of ADAR-driven viral restrictions in paramyxoviruses, orthomyxoviruses, flaviviruses, filoviruses, arenaviruses, and phenuiviruses are available [133]. In the measles virus (MV), belonging to the paramyxovirus family, A to I mutation in the matrix protein (M) by ADAR results in the introduction of stop codons and the subsequent decrease in M protein expression [135]. This low abundance of M protein hinders the viral nucleocapsids from colocalizing with the viral glycoproteins at the cell membrane. Consequently, virus nucleocapsids accumulate in the cell, restricting the release of infectious virus particles [136]. Similar to MV, other paramyxoviruses, such as the human respiratory syncytial virus (RSV) and parainfluenza virus 5 (PIV5), are also restricted by ADAR-driven biased hypermutations of different viral genes [131,137,138]. In HCV, ADAR-driven RNA modification was found to limit HCV replication by limiting protein kinase R (PKR)-mediated translational arrest [139]. PKR activity is beneficial for HCV replication, as PKR-mediated translational arrest selectively blocks 5′-cap-dependent translation of cellular mRNAs but does not affect internal ribosome entry site-dependent translation initiation of HCV RNA [140]. Interestingly, ADAR has been shown to restrict the Ebola virus (EBOV) in bats but not in humans. Bat cells constitutively express high levels of ADAR1, which causes hypermutations of the EBOV glycoprotein gene. However, such hypermutations are not observed when the virus is passaged in human embryonic kidney (HEK)-293T cells, which have lower ADAR1 expression [141]. ADAR1 p110-mediated RNA editing restricts influenza A virus (IAV) replication. Replication of different IAV strains was found to be five- to ten-fold higher in p110 KO cells compared to control [142]. The lymphocytic choriomeningitis virus (LCMV) of the Arenaviridae family and the Rift Valley fever virus (RVFV) of the Phenuiviridae family are also restricted by ADAR-mediated A to I mutations in their glycoprotein and RNA polymerase genes, respectively [143,144]. Similar A:I editing by ADAR1 has also been reported to antagonize the biogenesis of encephalomyocarditis virus (EMCV) circular RNAs [145]. Virus-associated RNA-I (VAI) of adenoviruses has been found to be a potent inhibitor of ADAR1 [146]. However, neither VAI nor any other AdV mRNA is known to be edited by ADARs. Thus, the purpose of this antagonism of ADAR1 by AdVs is not understood, and further research is needed to decipher the role of ADAR in AdVs. In contrast to these restriction factor roles of ADARs, in the Zika virus (ZIKV), ADAR1 has been reported to promote viral replication by inhibiting the activation of protein kinase PKR [147]. Similarly, in vesicular stomatitis virus (VSV), MV, and retroviruses, ADAR1 plays a proviral role by inhibiting PKR and by enhancing the expression of viral proteins [148,149,150,151,152,153].

#### 2.2.2. Pseudouridine Synthases (PUS)

Pseudouridine synthases (PUS) catalyze the isomerization of uridine to 5-ribosyl uracil, also known as pseudouridine (Ψ), which is the most abundant RNA modification in eukaryotes and is often referred to as the fifth nucleotide. Pseudouridine promotes incorporation of alternative amino acids during translation, leading to amino acid substitution, especially when aminoacyl-tRNAs are limited [154]. Although pseudouridine is a common RNA modification, it has only been identified in a few reports in the RNAs of eukaryotic viruses [129,155], and its function in regulating viral gene expression remains largely unexplored in mammals. 

PUS proteins were identified as potential antiviral candidates during a CRISPR screen for host factors targeting viruses in the Flaviviridae family—HCV and DENV [129,156]. Although they were not the primary antiviral factors identified in the screen, the identification of PUS proteins provides a basis for further studies on the role of pseudouridine in viral infections.

#### 2.2.3. N6-Methyladenosine (m6A) Writers, Readers, and Erasers

The methylation of adenosine residues (m6A) in nascent RNA is one of the most well-studied epitranscriptomic RNA modifications. The m6A machinery includes m6A writer, eraser, and reader proteins, which together regulate RNA splicing, RNA export, RNA degradation, and RNA translation. Two methyltransferase enzymes (writers), METTL3 (methyltransferase 3) and METTL14 (methyltransferase 14), together with WTAP (Wilms tumor 1-associated protein) and VIRMA (Vir Like M6A Methyltransferase-Associated) catalyze m6A modifications on nascent RNAs [157,158]. On the other hand, demethylating enzymes, FTO (fat mass and obesity-associated protein), and ALKBH5 (α-ketoglutarate-dependent dioxygenase AlkB homology 5) erase m6A marks [159]. Several m6A readers, such as YTHDF1, YTHDF2, YTHDF3, YTHDC1, YTHDC2, Pho92, and IGF2BP2, have been identified in eukaryotes that recognize m6A modifications and regulate RNA fate [160]. 

Like the RNA ‘editing’ mechanisms discussed above, the m6A RNA ‘modification’ mechanism has both proviral and antiviral effects. While there are more instances of m6A modification of viral RNAs having proviral effects, there are reports of m6A machinery acting as a viral restriction factor, albeit fewer in number. In Zika virus, a flavivirus, KD of the m6A writer METTL3 increased virus replication, while KD of the m6A eraser ALKBH5 (AlkB homolog 5) and FTO (fat mass and obesity-associated protein) exerted an inhibitory effect on viral replication [161,162]. In retroviruses, the human m6A reader YTHDF3 (YTH N6-Methyladenosine RNA-binding protein F3) has been reported to inhibit HIV-1 replication, albeit at a moderate level [163,164]. It has also been reported that HIV-1 proteases antagonize this restriction by degrading the YTHDF3 molecules associated with the virion [165]. Intriguingly, several other reports indicate that m6A modifications actually promote HIV-1 gene expression [166,167,168]. Therefore, the antiviral role of m6A in HIV-1 biology is controversial. However, a more defined restrictive role of m6A has emerged in ERVs. Depletion of METTL3 and METTL14, along with their accessory subunits WTAP and ZC3H13, resulted in increased mRNA abundance of mouse IAPs and related ERVK elements in mouse embryonic stem cells [169].

## 3. Future Directions

### 3.1. The Possibility of a Supramolecular ‘Restrictosome’

Among the four chromatin-targeting eRFs identified to date, two are dynamic macromolecular complexes: the PML-NBs and the KRAB/KAP1 complex. These complexes recruit numerous other epigenetic modulators like various HMTases, various HDACs, various DNMTs, EZH2, MDM2, HP1, MeCP2, MacroH2A2, SETDB1, RAD50, p300, etc., to silence the transcription of their targeted loci [63,170,171]. In addition to being antiviral restriction factors, both of these complexes are established as being important epigenetic modulators in the cellular milieu, having other physiological functions where their transcription repression function serves diverse roles such as in DNA damage response (DDR), DNA repair, telomere homeostasis, genome maintenance, cell division, cell proliferation, suppression of tumor growth, cellular senescence, and programmed cell death [63,172,173,174,175]. IFI16, on the other hand, is more established as an innate immune DNA sensor where it plays an indispensable role in the initial detection of foreign DNA in the nucleus leading to the induction of the inflammasome and the interferon responses. Though no specific enzymatic or epigenetic ability has been recognized for IFI16, it has been reported to recruit other epigenetic silencing proteins to transcriptionally restrict viral DNA. However, whether IFI16 also forms an intricate and dynamic macromolecular complex remains to be convincingly elucidated. Interestingly, a recent study reported that IFI16 interacts with KAP1 and SZF1, a KRAB-ZFP, to silence the EBV lytic switch protein ZEBRA, encoded by the BZLF1 gene [74]. This partnership between these two seemingly independent epigenetic machineries blurs the individuality of these systems and invokes the question of whether they together form an even larger supramolecular complex. More intriguingly, the authors of another study reported that the nuclear filamentous structures of IFI16 that form on viral DNA interact with and recruit PML-NBs resident proteins, including PML, Sp100, and ATRX [91]. They also observed that the ability to form IFI16 filaments in different cell types correlates with the efficiency of epigenetic restriction. These authors speculated that IFI16 might serve as a recruiting platform for other restriction factors like ATRX and has the capability to exert restrictive functions both in cis and trans [91]. Other studies also reported that IFI16 forms aggregate or fibrous structures in the nuclei of infected cells [176,177,178] and that IFI16 interacts with the PML-NB proteins [179]. Another study that explored IFI16 interacting proteins by proteomic approaches also identified PLM and SP100 as IFI16 interactors [177]. In addition, this study also reported that IFI16 is targeted to PML-NBs following infection with an ICP0 mutant HSV-1. On the other hand, the HUSH complex has been reported to cooperate with KAP1 to repress the transcription of retrotransposons [121]. However, this study did not report any physical interaction between the HUSH complex and KAP1. Many epigenetic effector proteins are common factors between eRFs. For example, the H3K9me3 MTase, SETDB1, is the major H3K9 methylating effector for the PML-NBs, the KRAB/KAP1 complex, and the HUSH complex. Similarly, HDACs and HP1α are also common to multiple eRF systems. 

Taken together, these independent observations suggest that all four chromatin-targeting eRFs discovered to date, PML-NBs, KRAB/KAP1, IFI16, and the HUSH complex interact with each other and possibly coordinate their silencing effects in the form of a nuclear supramolecular complex. In agreement with this hypothesis, Leiberman and colleagues have proposed the term “restrictosome” to describe the IFI16 filaments and their associated restriction factors in the nucleus of infected ICP0 null HSV-1 cells [91]. It is tempting to speculate that these “restrictosomes” encompass not only IFI16 and its interactors, but rather, some other eRFs and their effectors as well. The persisting question that remains to be answered is how comprehensive or dynamic these supramolecular structures are and what other epigenetic machinery they contain. 

However, some observations also argue against the presence of an universal supramolecular restrictosome for silencing all invading viral DNA. First, soluble forms of some PML-NB resident proteins have been reported to be able to mediate epigenetic restriction [54]. ATRX, a PML-NB resident protein, has been reported to be only partially localized to the IFI16 filaments, with both, soluble as well as IFI16 independent ATRX puncta visible inside infected cells [91]. Second, the filamentous assemblies of IFI16 were observed to be associated with only a subset of the ICP0 null HSV1 genomes inside the nucleus of infected cells; however, epigenetic restriction and reduction in RNA-Pol II recruitment were observed on all the HSV1 genomes present, even those not associated with the IFI16 supramolecular assemblies [91]. The authors speculated that the IFI16 filamentous assemblies function as a signaling platform that senses invading viral DNA molecules and signals in trans to silence all the viral DNA molecules inside the nucleus. Third, another study investigating whether IFI16 and ND10 components work together or separately to restrict HSV-1 replication concluded that the mechanisms of action of IFI16 and PML-NB proteins are independent, at least in part [180]. Thus, more research is needed to understand the crosstalk and association between the different chromatin-targeting eRFs conclusively. 

A recent report has stated that the restriction factor APOBEC3A (Apolipoprotein B mRNA-editing enzyme catalytic polypeptide-like 3G) recruits KAP1 and, subsequently, HP1 and H3K9me3 to suppress HIV-1 transcription, helping in maintaining HIV-1 latency [181]. Overexpression of APOBEC3A in latently infected cells led to lower reactivation, while KD or KO of APOBEC3A led to increased spontaneous and inducible HIV-1 reactivation [181]. APOBEC3A is a cytidine deaminase that deaminates cytosine residues to uracil, leading to a cytosine-to-thymine (C:T) mutation, which is lethal for viruses like HIV-1, HBV, HPV, EBV, HCMV, and HSV1 [182,183,184,185]. However, it is not an epigenetic modulator. The finding that APOBEC3A associates with the KAP1-mediated epigenetic restriction mechanism is unexpected and prompts further investigation of the possible crosstalk between these discrete viral restriction mechanisms. In humans, APOBECs can also deaminate the epigenetic DNA mark, 5-methylcytosine (5mC) to thymine, creating T:G mismatches, which, followed by error-free DNA repair, leads to mutations [186,187,188]. However, to date, this 5mC deaminating activity has not been described in the restriction of any virus, and the mutation of 5mC:T is not an epigenetic modification. For these reasons, APOBECs themselves cannot be considered as eRFs, but they may interact with and recruit other eRFs in addition to KAP1.

### 3.2. Crosstalk between Epitranscriptomics and Epigenetics

Recent reports of crosstalk between epitranscriptomics and epigenetics have opened new frontiers in our understanding of gene regulation. In addition to regulating the fate of RNA transcripts, the epitranscriptomic machinery has also been found to affect gene expression by modulating epigenetic marks on corresponding genomic loci [189,190,191,192]. Often, this forms a critical epigenetic checkpoint via which the RNA transcript regulates the expression of its cognate gene. Research in mice has shown that m6A can impact numerous epigenetic modifications, including H3K4me3, H3K27ac, H3K27me3, and H3K9me2/3 (comprehensively reviewed in [189]). In one example of such cross-regulation, the m6A readers YTHDC1 (YTH N6-Methyladenosine RNA Binding Protein C1) was shown to interact with and recruit the H3K9 demethylase KDM3B to demethylate H3K9me2 co-transcriptionally [191]. A separate report demonstrated that YTHDC1 plays a crucial role in facilitating the interaction between the m6A methyltransferase METTL3 and the mouse chromatin with the epigenetic factors SETDB1 and KAP1. This, in turn, leads to the co-transcriptional deposition of H3K9me3 on mouse ERVs such as IAPs [193]. Other studies have reported similar links between m6A and heterochromatin formation in mice [169,194,195]. In addition to the regulation of epigenetics by epitranscriptomic mechanisms, the reciprocal has also been reported where epigenetic machinery drives change in epitranscriptomics. In one such example, the H3K36me3 mark was found to drive the binding of the m6A methyltransferase complex (MTC) to the nascent transcripts to co-transcriptionally methylate them [196]. 

Viral transcription is a tightly regulated process, and such feedback mechanisms, in theory, should be beneficial for the precise temporal expression of different viral gene classes. However, no reports of such crosstalk between viral epitranscriptomics and epigenetics exist, but the possibility that such regulatory networks may exist in viruses is nonetheless exciting.

### 3.3. Epidrugs and the Potential of Targeting eRFs

As more knowledge of different epigenetic mechanisms is discovered, epidrugs (epigenetic and epitranscriptomic drugs) are gaining prominence in treating human diseases, ranging from cancer to neurodegenerative conditions [197,198]. Epigenetic drugs can be classified into three broad groups: epigenetic ‘writer’ inhibitors, which include DNA methyltransferase inhibitors (DNMTi), histone methyltransferase inhibitors (HMTi), and histone acetyltransferase inhibitors (HATi). Epigenetic ‘eraser’ inhibitors include histone demethylase inhibitors (HDMi) and histone deacetylase inhibitors (HDACi). And, epigenetic ‘reader’ inhibitors, such as bromodomain and extra-terminal motif protein inhibitors (BETi), ten-eleven translocation inhibitors (TETi), and chromatin remodeler inhibitors (CRi). To date, eight epigenetic drugs have been approved by the USA FDA, mostly for the treatment of cancer [199]. In comparison to epigenetic drug discovery, targeting the epitranscriptome is still in its early stages [198]. This includes RNA methyltransferase inhibitors (RNMTi) and RNA demethylase inhibitors (RNDMi). Methyl-RNA-reader inhibitors, though theoretically feasible, have not been developed yet.

The use of epidrugs for the treatment of viral diseases has enormous potential. However, epidrugs can also affect the regulation of host genes, which may result in unintended side effects. So far, no specific antiviral epigenetic drug has been developed. Nevertheless, repurposing epigenetic drugs originally developed to treat other diseases, such as cancer, has been successful [200]. As previously discussed, many viruses utilize specific host eRFs to orchestrate their lifecycle, especially latency establishment and maintenance. A number of the epigenetic inhibitors developed to date target molecules that are part of these eRF complexes. For example, inhibitors of the PML-NB-associated H3K27me3 ‘writer’ EZH2, which were originally developed for the treatment of various cancers [25], have been found to suppress HSV, HCMV, and AdV gene expression and lytic replication in cell culture [201]. In the case of HIV, combination antiretroviral therapy (cART), though very effective, is limited by its inability to target integrated latent proviruses. Epigenetic drugs targeting host chromatin-modifying enzymes essential for the transcriptional silencing of these proviruses have led to the “shock and kill” strategy against latent HIV. In this strategy, the idea is to use epigenetic drugs to induce the latent provirus into an active form (shock), which then can be targeted by the humoral immune response, CD8+ T cell-mediated lysis, virus-induced apoptosis, or activation-induced cell death. Several latency-reversing agents (LRA) or “shock” inducers have been proposed, including HDACi, HMTi, and DNMTi (comprehensively reviewed previously in [200]. Similar “shock and kill” approaches using epigenetic drugs have been found to be effective against HCMV, KSHV, and EBV [200,202,203].

Several epigenetic drugs are undergoing clinical trials for the treatment of viral diseases. These include panobinostat (ClinicalTrial.gov #NCT01680094, HIV), vorinostat (#NCT01319383, HIV), romidepsin (#NCT02092116, #NCT01933594, HIV), VPA (#NCT00289952, HIV), belinostat (#NCT00321594, HBV and HCV), tractinostat (#NCT03397706, EBV), nanatinostat (#NCT05011058, #NCT05166577, EBV).

## 4. Concluding Remarks

Epigenetic silencing of viral nucleic acids is critical for eukaryotic cells to minimize viral gene expression and prevent the hostile takeover of cellular machineries by the invading virus. eRFs are those intrinsic cellular defense proteins that recognize, target, and silence viral gene promoters, or mRNAs. As per our current understanding, there is not much difference between viral chromatin or mRNAs from that of the host. This makes the viral chromatin and mRNAs susceptible to all the gene regulatory networks functional in the host. However, to date, only a handful of elements of cellular epigenetic machinery have been identified to be functional in viruses. There is a fair possibility that future research in this field will uncover more roles of cellular epigenetic machineries in viral lifecycles.

## Figures and Tables

**Figure 1 viruses-16-00183-f001:**
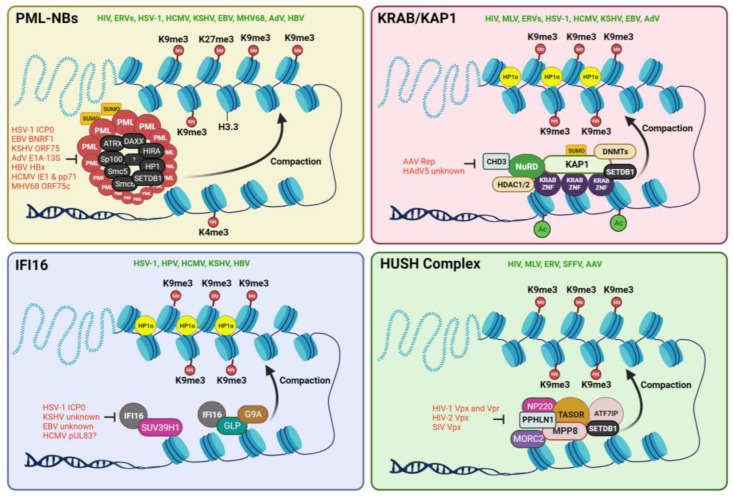
Four different chromatin-targeting eRFs have been identified to date. These are the PML-NBs, the KRAB ZNF/KAP1 complex, the IFI16, and the HUSH complex. These eRFs recruit other epigenetic modifiers that help recruit repressive histone marks like H3K9me3 on the genomes of target viruses. This results in heterochromatinization of the viral chromatin, making them more compact and inaccessible to RNA polymerases and other transcriptional machineries. Each of these eRFs has been reported to epigenetically restrict several different viruses in the literature (denoted in green). Additionally, some viruses have developed strong antagonistic mechanisms to prevent the activity of eRFs during infection or reactivation from latency. These antagonistic viral proteins are illustrated in red. The chromatin in blue denotes viral genomes, either episomal or integrated. Illustration created with BioRender.com.

**Figure 2 viruses-16-00183-f002:**
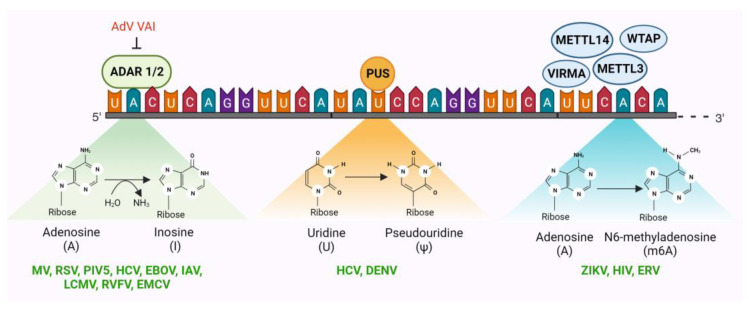
Three epitranscriptomic systems have been identified to serve as antiviral restriction factors. Two among these, ADAR (adenosine deaminase acting on RNA) and PUS (pseudouridine synthases), are RNA-editing mechanisms that introduce mutations in the reading frame post-transcriptionally. The third, N6-methylasenosine writers (METTL3, METTL14, WTAP, and VIRMA), readers, and erasers, together modify viral RNA and determine its intracellular fate. Each of these epitranscriptomic eRFs has been reported to restrict several viruses in the literature (denoted in green). Adenovirus (AdV) virus-associated RNA-1 (VAI) functions as a potent inhibitor of ADAR1 but not ADAR2. Considering the fact that neither the VAI nor any other AdV mRNA is a substrate for ADAR1, the function of this antagonism is unknown. Illustration created with BioRender.com.

## Data Availability

Not applicable.

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
