# Peer review of "Epigenetic Restriction Factors (eRFs) in Virus Infection"

_viruses, 2024, doi:10.3390/v16020183_

Round 1

Reviewer 1 Report

Comments and Suggestions for Authors

The manuscript on epigenetic restriction factors (eRFs) was excellently written and contains with a very comprehensive and updated description of different biochemical processes that lead to viral restriction. The figures are very neat and appropriate to illustrate both major mechanisms of restriction:  the one that targets viral chromatin and the other one that affects viral RNA.

I have two minor points that the authors may consider in their revised version:

1) For those molecular players that bind to viral DNA/chromatin such as PML-NBs or ND10, KRAB/KAP1, HUSH, and others, what is known about sequence specificity? Do they bind to any specific DNA sequence? Do they recognize any secondary structure? This point will clarify the issue of how these eRFs distinguish self versus non-self genetic elements. 

2) It would be interesting to discuss how these molecular mechanisms on viral restriction based on cellular epigenetic players can be applied in the clinic to treat viral infections. I recommend adding some paragraphs including (if there are) clinical trials/studies on epigenetic modifiers and infectious diseases. 

Author Response

Reviewer#1

The manuscript on epigenetic restriction factors (eRFs) was excellently written and contains with a very comprehensive and updated description of different biochemical processes that lead to viral restriction. The figures are very neat and appropriate to illustrate both major mechanisms of restriction:  the one that targets viral chromatin and the other one that affects viral RNA.

I have two minor points that the authors may consider in their revised version:

  • For those molecular players that bind to viral DNA/chromatin such as PML-NBs or ND10, KRAB/KAP1, HUSH, and others, what is known about sequence specificity? Do they bind to any specific DNA sequence? Do they recognize any secondary structure? This point will clarify the issue of how these eRFs distinguish self versus non-self genetic elements. 

--> We thank the reviewer for the suggestion. New information on the sequence specificity of all the eRFs discussed is now included in the revised manuscript (lines 203-204, 237-241, 324-326, 365-366).  

  • It would be interesting to discuss how these molecular mechanisms on viral restriction based on cellular epigenetic players can be applied in the clinic to treat viral infections. I recommend adding some paragraphs including (if there are) clinical trials/studies on epigenetic modifiers and infectious diseases. 

--> We appreciate this recommendation and have now included a new section (Section 3.3) discussing ‘Epidrugs and the potential of targeting eRFs’ (lines 641-682).

Reviewer 2 Report

Comments and Suggestions for Authors

Restriction factors (RFs) are a frontline defense against viral infection, the host cellular proteins that target specific stages of the viral life cycle to limit viral replication and are often essential for slowing viral replication in vivo until an adaptive immune response can further reduce or eliminate infection. In fact, RFs generally target viral features or replication mechanisms that are highly conserved within viral orders or families. They constitute the basis of intrinsic antiviral immunity, which is considered either part of the innate immune response or an independent third branch of the immune system.

This manuscript describes two categories of cellular restriction factors, which are part of the epigenetic machinery, and presents organized information and insights on epigenetic restriction factors (eRFs) that are important for combating viral infection and virus-induced disease. In general, the manuscript is well written and can be published in Viruses.

-The main concern that can be raised against such a review is whether this type of classification covers all eRFs and/or relevant mechanisms.  For example, LSD1/KDM1A is a widely conserved lysine-specific demethylase that removes methyl groups from methylated proteins, mainly histone H3, which plays an important role in mediating the expression of genes involved in viral infection. How can this be classified and included in these two categories?

- The protein complexes in these two categories can interact? Crosstalk between epitranscriptomics and epigenetics need further discussion, I suggest to provide more details and if possible, via a good illustration, beyond a part of the Conclusions and Future Directions section!

- Referencing is another problem with this manuscript, lines 29-38, 66-77 and ... Figure 1 is also not referenced in the text.

Author Response

Reviewer#2

Restriction factors (RFs) are a frontline defense against viral infection, the host cellular proteins that target specific stages of the viral life cycle to limit viral replication and are often essential for slowing viral replication in vivo until an adaptive immune response can further reduce or eliminate infection. In fact, RFs generally target viral features or replication mechanisms that are highly conserved within viral orders or families. They constitute the basis of intrinsic antiviral immunity, which is considered either part of the innate immune response or an independent third branch of the immune system.

This manuscript describes two categories of cellular restriction factors, which are part of the epigenetic machinery, and presents organized information and insights on epigenetic restriction factors (eRFs) that are important for combating viral infection and virus-induced disease. In general, the manuscript is well written and can be published in Viruses.

  • The main concern that can be raised against such a review is whether this type of classification covers all eRFs and/or relevant mechanisms.  For example, LSD1/KDM1A is a widely conserved lysine-specific demethylase that removes methyl groups from methylated proteins, mainly histone H3, which plays an important role in mediating the expression of genes involved in viral infection. How can this be classified and included in these two categories?

--> In DNA viruses, KDM1A plays a pro-viral role as inhibition or knockdown of KDM1A blocks viral genome transcription and replication (Mol Ther. 2022 Jun 1;30(6):2153-2162, and Antimicrob Agents Chemother. 2014 May; 58(5): 2807–2815.). Therefore, KDM1A is not a restriction factor (RF) for DNA viruses. In RNA viruses, KDM1A has been reported to limit virus replication by demethylating and activating the cholesterol homeostasis-maintaining protein IFITM3 (Interferon-Induced Transmembrane Protein 3) at position K88, which, in turn, restricts the entry of many RNA viruses (PLoS Pathog. 2017 Dec; 13(12): e1006773). This mechanism is not epigenetic. Therefore, LSD1/KDM1A can not be considered as an epigenetic restriction factor (eRF), which is the topic of this review. Similarly, there are other epigenetic/ non-epigenetic factors that have been reported to regulate viral replication in the literature but have not been classified as eRFs in this review due to reasons that are discussed in the manuscript (lines 143-151 and 605-614).

  • The protein complexes in these two categories can interact? Crosstalk between epitranscriptomics and epigenetics need further discussion, I suggest to provide more details and if possible, via a good illustration, beyond a part of the Conclusions and Future Directions section!

--> We appreciate the reviewer’s suggestion. Accordingly, section 3.2 - “Crosstalk between epitranscriptomics and epigenetics” has been rewritten with more details and additional references. We also agree that an illustration can be helpful in this situation, but there is already an excellent review of this topic available with appropriate illustrations in the February 2022 issue of Trends in Genetics (volume 38, issue 2, pages 182-193). To avoid redundant information, we have suggested referring to this source for more details in our manuscript.

  • Referencing is another problem with this manuscript, lines 29-38, 66-77 and ... Figure 1 is also not referenced in the text.

--> We thank the reviewer for noticing this lapse on our part. We have now incorporated appropriate references in the suggested sections and referred to Fig 1 in the text.

Reviewer 3 Report

Comments and Suggestions for Authors

The manuscript of  Arunava Roy and Anandita Ghosh investigates eRFs and their role in limiting viral infections, and the mechanisms by which
viruses have evolved to counteract them. Moreover, they examine the crosstalk between different eRFs, including the common effectors that connect them. Finally, they explore the potential for new discoveries in the realm of epigenetic networks that restrict viral gene expression.

The subject is very interesting and the manuscript is a well written review. The main topic is to discuss the eRFs mechanism of restriction, interacting partners, and their effect on viral transcription and replication.Refeferences are updated and well discussed.

Figures 1 and 2 are well presented summarizing the different eRFs that are described in the manuscript.

Minor suggestions:

1) It will be of interest to include a summary information in a figure about eRFs that target viral chromatin and epitranscriptomics eRFs in  different viruses.

2) A final figure summarizing the parts 3.1 and 3.2 and their putative involvement  in viral infections   will be an added value to the manuscript.

Author Response

Reviewer#3

The manuscript of  Arunava Roy and Anandita Ghosh investigates eRFs and their role in limiting viral infections, and the mechanisms by which viruses have evolved to counteract them. Moreover, they examine the crosstalk between different eRFs, including the common effectors that connect them. Finally, they explore the potential for new discoveries in the realm of epigenetic networks that restrict viral gene expression.

The subject is very interesting and the manuscript is a well written review. The main topic is to discuss the eRFs mechanism of restriction, interacting partners, and their effect on viral transcription and replication.Refeferences are updated and well discussed.

Figures 1 and 2 are well presented summarizing the different eRFs that are described in the manuscript.

Minor suggestions:

  • It will be of interest to include a summary information in a figure about eRFs that target viral chromatin and epitranscriptomics eRFs in  different viruses.

--> We thank the reviewer for this suggestion. We have modified Figs 1 and 2 to include all the different viruses (green text) reported in the literature as restricted by the corresponding eRF.

  • A final figure summarizing the parts 3.1 and 3.2 and their putative involvement in viral infections will be an added value to the manuscript.

--> We appreciate the reviewer’s recommendation; however, due to its hypothetical nature, an illustration for section 3.1 is difficult to formulate and could possibly be misleading to the reader. Section 3.2 is more established, but there is already an excellent review of this topic available with appropriate illustrations in the February 2022 issue of Trends in Genetics (volume 38, issue 2, pages 182-193). To avoid redundant information, we have suggested referring to this source for more details.